# Phenotypic and Genetic Analyses of In Vitro Embryo Production Traits in Chinese Holstein Cattle

**DOI:** 10.3390/ani13223539

**Published:** 2023-11-16

**Authors:** Yuechuan Huang, Hailiang Zhang, Cheng Mei, Minglu Yang, Shanjiang Zhao, Huabin Zhu, Yachun Wang

**Affiliations:** 1State Key Laboratory of Farm Animal Biotech Breeding, National Engineering Laboratory for Animal Breeding, Key Laboratory of Animal Genetics, Breeding and Reproductive of Ministry of Agriculture and Rural Affairs, College of Animal Science and Technology, China Agricultural University, Beijing 100193, China; hyccau@cau.edu.cn (Y.H.); zhl108@cau.edu.cn (H.Z.); 2College of Animal Science and Technology, Xinjiang Agricultural University, Urumqi 830052, China; 3Dongying Auatasia Modern Animal Husbandry Co., Ltd., Dongying 257300, China; cheng.mei@austasiadairy.com (C.M.); minglu.yang@austasiadairy.com (M.Y.); 4Institute of Animal Science, Chinese Academy of Agricultural Sciences, Beijing 100193, China; zhaoshanjiang@caas.cn

**Keywords:** in vitro embryo production, Holstein cattle, phenotypic analysis, genetic parameters

## Abstract

**Simple Summary:**

Ovum pick up and in vitro embryo production (OPU-IVEP) is a critical technique in the dairy industry that is attracting a lot of attention, especially in China’s dairy industry. The aim of the technique is to produce as many embryos with excellent genetic material as possible. However, there is a large variation in the ability of donor cows to produce embryos, and this ability is heritable. In this case, genetic selection and animal breeding will be beneficial to improving efficiency. Considering the large differences among different populations, the genetic parameters regarding the Chinese population are currently essential for genetic selection in the future. In this study, we analyzed the phenotypic and genetic characteristics of in vitro embryo production traits in Chinese Holstein cattle. The results of this study may help to understand the breeding potential of in vitro embryo production traits.

**Abstract:**

Ovum pick up and in vitro embryo production (OPU-IVEP) is an essential technique in the dairy industry. The production efficiency of OPU-IVEP is significantly influenced by various factors, and phenotypic and genetic characteristics are highly variable in different populations. The objectives of this study were (1) to reveal the phenotypic characteristics, including population distribution, and impacts of donor age and month on in vitro embryo production and (2) to estimate genetic parameters for five in vitro embryo production traits in Chinese Holstein cattle. A total of 7311 OPU-IVEP records of 867 Holstein heifers from August 2021 to March 2023 were collected in this study. Five in vitro embryo production traits were defined, including the number of cumulus–oocyte complexes (NCOC), the number of cleaved embryos (NCLV), the number of grade I embryos (NGE), and the proportion of NCLV to NCOC (PCLV) and NGE to NCOC (PGE). A univariate repeatability animal model was employed to estimate heritability and repeatability, and a bivariate repeatability animal model was employed to estimate the genetic correlations among five in vitro embryo production traits. It was found that the in vitro embryo production traits were significantly influenced by season, as the NGE and PGE were significantly decreased from June to August. In addition, the production efficiency of OPU-IVEP was also influenced by donor age. On the observed scale, the estimates of heritability were 0.33 for NCOC, 0.24 for NCLV, 0.16 for NGE, 0.06 for PCLV, and 0.10 for PGE, respectively. On the log-transformed scale, the estimates of heritability of NCOC, NCLV, and NGE were 0.34, 0.18, and 0.13. The genetic correlations among NCOC, NCLV, and NGE ranged from 0.61 (NCLV and NGE) to 0.95 (NCOC and NCLV), considering both scales. However, there were low genetic correlations between NCOC and proportion traits (PCLV and PGE) on both the observed scale and the log-transformed scale. In the end, the variation in Chinese Holstein cattle was found to be considerable. The EBV value and average NCOC, NGE, and PGE for the top 10% donors presented extreme differences to those for the bottom 10% donors for NCOC (24.02 versus 2.60), NGE (3.42 versus 0.36), and PGE (30.54% versus 3.46%). Overall, the results of this study reveal that in vitro embryo production traits are heritable with low to high heritability, and the count traits (NCOC, NCLV, and NGE) and proportion traits (PCLV and PGE) reflect different aspects of in vitro embryo production and should be incorporated into genetic selection for improving the embryo production efficiency of dairy cattle.

## 1. Introduction

In recent decades, assisted reproductive technologies have been critical tools in domestic industry [1]. In vitro fertilization (IVF), in vitro embryo production (IVEP), and embryo transfer (ET) have allowed superior females to produce more offspring and have also played an innovative role in genetic improvement, helping to maintain a low inbreeding rate [2,3], which is of great importance for dairy cattle breeding. According to statistics from the International Embryo Technology Society (www.iets.org, accessed on 1 December 2022), a total of 1,499,136 bovine embryos were produced in vitro in 2021, almost four times the number produced in vivo. However, there was high variability in the embryo production efficiency of cows, and large differences were found for the number of in vitro embryos produced by each donor cow. To increase efficiency and improve the benefit of in vitro embryo production, identifying and selecting proper donors prior to the ovum pick up (OPU) operation is the next step [4].

The technique firstly relies on the ultrasound-guided transvaginal ovum pick up operation to collect cumulus–oocyte complexes (COC). The number of COC (NCOC) are similar to the number of viable oocytes, which is also a commonly used trait. The NCOC is the first trait used to evaluate donor ability [1]. After 24 h of in vitro maturation, the oocytes are subjected to in vitro fertilization and the number of cleaved embryos (NCLV) is recorded. Then the embryos remain in culturing for 7 days. Subsequently, the embryos are evaluated and categorized. The number of grade I embryos (NGE), similar as number of good/viable embryos, are recorded. The NGE is considered an important trait that characterizes profitability directly [1]. Although fertilization can affect the NCLV and NGE, several studies have found that the growth process of oocytes is essential to the outcome of embryo maturation [5]. In addition, percentage traits are ignored, and the traits reflect the donor’s ability to produce high-quality oocytes.

The effects of some environmental and physiological factors on in vitro embryo production in Holstein cattle and water buffalo have been reported in previous studies. For example, Zeron et al. [6] found that the number of oocytes per ovary in Holstein cattle in spring and winter was significantly higher than that in summer and autumn. The IVEP traits showed a similar trend in water buffalo, with the exception of the number of cumulus–oocyte complexes [7]. Generally, poor OPU performances were observed in summer, and operation season partly explains the phenotypic variation in embryo production efficiency [8,9]. In addition, the effects of other factors, such as donor age, donor parity and physiological status, diet, and gonadotropin stimulation, on oocyte collection and embryo production have also been noted in a few reports [7,10,11,12]. In China, dairy cattle are challenged by serious heat stress in summer [13]. However, its effect on embryo production in Chinese dairy populations has not been well reported.

Except for the non-genetic factors previously mentioned, differences in genetic background are further reasons for the variation in IVEP [14,15,16]. In a few previous studies, the genetic architecture of IVEP traits has been investigated in dairy Gir cattle [17], Guzera cattle [18], and Holstein cattle [2,19]. In vitro embryo production traits are considered to be heritable, and the estimated heritability of IVEP traits ranges from 0.15 to 0.34. The heritabilities of IVEP traits present high levels of variation across populations. In addition, genetic correlations have not always been investigated, and the estimates of genetic correlations are also considerable [14,17,19]. The genetic characteristics between different populations are inconsistent, and more studies are needed.

The development of IVEP technology has attracted increasing interest, and investigating the genetic architecture of IVEP traits is important for improving embryo production efficiency. IVEP technology has also developed rapidly in the Chinese dairy industry, and IVEP traits are useful for donor selection and expected to be introduced into breeding projects in the future. However, to our knowledge, there is a lack of studies focusing on the genetic parameters of IVEP traits in Chinese dairy cattle. The objectives of the current study were to reveal the population characteristics of IVEP traits in Chinese Holstein cattle, including phenotypic and genetic parameters. The present study contributes to the genetic basis of cow reproductive performance and provides useful information for achieving breeding objectives in cattle.

## 2. Materials and Methods

### 2.1. Animal and Data

In this study, a total of 7311 OPU-IVEP records were collected. These records were from 867 Holstein heifers, daughters of 70 sires and 387 dams, and OPU operations were performed from August 2021 to March 2023 in the Austasia herd (Shandong, China). The age of the donors ranged from 8 to 37 months, and the majority of donors were between 9 and 19 months of age (98.9%), with good milk production performance. The average frequency of OPU operation per donor cow was 8.4 times.

The pedigree was provided by the Dairy Association of China (Beijing, China). Animals with OPU records were traced for as many generations as possible. Finally, the pedigree file used in this study included 2919 females and 535 bulls born between 1945 and 2022.

### 2.2. Ovum Pick Up and In Vitro Embryo Production

Oocytes were collected via ultrasound-guided transvaginal ovarian puncture, which was carried out as described by Seneda [20]. The ovum pick up operations were performed by technicians using an IMV ECM-EXAPADE MINI ultrasound scanner with an 8-MHz microconvex array transducer and 18 G catheters with a vacuum pressure of 100 mm/hg. After the OPU, 2.5 mL tubes containing the structures collected from the donors were immediately transferred to a nearby laboratory. The solution containing the structures was then passed through an EmCom filter with phosphate-buffered saline and washed three times in a Petri dish with the same solution used in OPU operation.

The oocytes used for IVEP were incubated for 24 h at 38.5 °C under an atmosphere of 5% CO_2_ in air. After in vitro maturation (IVM), the oocytes were washed in fertilization medium and then incubated for 8 h in a Petri dish containing 100 µL microdrops of Tyrode’s albumin lactate pyruvate (TALP) medium supplemented with 10 μg/mL heparin and 160 µL penicillamine–hypotaurine–epinephrine. In vitro culturing (IVC) was performed for 8–10 h at 38.5 °C in an atmosphere of 5% CO_2_ in air. The culture medium was renewed for each microdrop after 2 days, and at 7 days after in vitro fertilization, the embryos were evaluated before freezing or transferred using the standard developed by Lonergan et al. [21]. The service sire, the number of cumulus–oocyte complexes (NCOC), the number of cleaved embryos (NCLV), and the number of grade I embryos (NGE) were recorded for every donor in operations.

### 2.3. Trait Definition and Transformation

Five IVEP traits were analyzed in this study, including NCOC, NCLV, NGE, PCLV, and PGE. The NCOC, NCLV, and NGE were defined as the number of all oocytes flushed in OPU operations, embryos cleaved after IVF, and grade I embryos evaluated at the end of IVC, respectively. The PCLV and PGE were defined as the proportion of NCLV to NCOC (PCLV) and NGE to NCOC (PGE), respectively.

To achieve normal distribution, the NCOC, NCLV, and NGE were transformed as follows:$$Y=\log_{10}\left(X+1\right)$$
where X represents NCOC, NCLV, and NGE on the observed scale; Y represents the corresponding traits on the log-transformed scale. As shown in Appendix A, normality tests were performed using SPSS 21.0 (IBM Corp., Armonk, NY, USA, 2012) to create Q_Q charts on observed scale and log-transformed scale.

### 2.4. Statistics

Variance and co-variance components for five IVEP traits were estimated using the average information restricted maximum likelihood algorithm implemented in the DMU software [22]. A univariate repeatability animal model was used to estimate heritability. The model fitted for the in vitro embryo production traits is as follows:(1)  y=Xβ+Za+Wpe+e
where y is the vector of observations for NCOC on the observed and log-transformed scales; β is the vector of fixed effects, including donor age (≤9, 10, 11, 12, 13, 14, 15, 16, 17, 18, 19, ≥20) and year/month of OPU operation (a total of 19 levels from August 2021 to March 2023); a is the vector of the random additive genetic effect of the donors; pe is the random permanent environmental effect of the donors; e is the random residual effect of the donors; and X, Z, and W are the incidence matrices connecting β, a, and pe.
(2)y=Xβ+Za+Mpat+Wpe+e
where y is the vector of observations for traits such as NCLV and NGE on the observed and log-transformed scales, as well as PCLV and PGE; β is the vector of fixed effects, including donor age and year/month of OPU operation; pat is the vector of random effect of the service sire; and M is the incidence matrix connecting pat. Other effects were the same as in model 1.

It was assumed that $a\sim N(0,A\sigma_{a}^{2})$, $pat\sim N(0,I\sigma_{pat}^{2})$, $pe\sim N(0,I\sigma_{pe}^{2})$ and $e\sim N(0,I\sigma_{e}^{2})$, where A is the matrix of additive genetic relationships between individuals in the pedigree; **I** is the identity matrix; and $\sigma_{a}^{2}$, $\sigma_{pat}^{2}$, $\sigma_{pe}^{2}$, and $\sigma_{e}^{2}$ are the additive genetic variance, paternal random variance, permanent environmental variance, and residual variance of the IVEP traits, respectively.

To estimate the genetic correlations between NCOC and other traits, a bivariate repeatability animal model was constructed, and this model is as follows:(3)$$\left[\begin{array}{c} y_{1}\\ y_{j} \end{array}\right]=\left[\begin{array}{cc} X_{1} & 0\\ 0 & X_{j} \end{array}\right]\left[\begin{array}{c} \beta_{1}\\ \beta_{j} \end{array}\right]+\left[\begin{array}{cc} Z_{1} & 0\\ 0 & Z_{j} \end{array}\right]\left[\begin{array}{c} a_{1}\\ a_{j} \end{array}\right]+\left[\begin{array}{cc} M_{1} & 0\\ 0 & M_{j} \end{array}\right]\left[\begin{array}{c} 0\\ {pat}_{j} \end{array}\right]+\left[\begin{array}{cc} W_{1} & 0\\ 0 & W_{j} \end{array}\right]\left[\begin{array}{c} {pe}_{1}\\ {pe}_{j} \end{array}\right]+\left[\begin{array}{c} e_{1}\\ e_{j} \end{array}\right]$$
where $y_{1}$, $\beta_{1}$, $a_{1},\;$ and ${pe}_{1}$ are the same as in model 1, while $y_{j}$, $\beta_{j}$, ${a_{j}}_{j}$, ${pat}_{j}$, and ${pe}_{j}$ are the same as in model 2.

To estimate the genetic correlations among other traits, another bivariate repeatability animal model was constructed, and this model is as follows:(4)$$\left[\begin{array}{c} y_{i}\\ y_{j} \end{array}\right]=\left[\begin{array}{cc} X_{i} & 0\\ 0 & X_{j} \end{array}\right]\left[\begin{array}{c} \beta_{i}\\ \beta_{j} \end{array}\right]+\left[\begin{array}{cc} Z_{i} & 0\\ 0 & Z_{j} \end{array}\right]\left[\begin{array}{c} a_{i}\\ a_{j} \end{array}\right]+\left[\begin{array}{cc} M_{i} & 0\\ 0 & M_{j} \end{array}\right]\left[\begin{array}{c} {pat}_{i}\\ {pat}_{j} \end{array}\right]+\left[\begin{array}{cc} W_{i} & 0\\ 0 & W_{j} \end{array}\right]\left[\begin{array}{c} {pe}_{i}\\ {pe}_{j} \end{array}\right]+\left[\begin{array}{c} e_{i}\\ e_{j} \end{array}\right]$$

The fixed and random effects included in this model were the same as in model 2. It was assumed that $\left[\begin{array}{c} a_{i}\\ a_{j} \end{array}\right]\sim N(0,A⨂\left[\begin{array}{cc} \sigma_{a_{i}}^{2} & \sigma_{a_{i}a_{j}}\\ 0 & \sigma_{a_{j}}^{2} \end{array}\right]$, $\left[\begin{array}{c} {pat}_{i}\\ {pat}_{j} \end{array}\right]\sim N(0,I⨂\left[\begin{array}{cc} \sigma_{{pat}_{i}}^{2} & \sigma_{{pat}_{i}{pat}_{j}}\\ 0 & \sigma_{{pat}_{j}}^{2} \end{array}\right]$, $\left[\begin{array}{c} {pe}_{i}\\ {pe}_{j} \end{array}\right]\sim N(0,I⨂\left[\begin{array}{cc} \sigma_{{pe}_{i}}^{2} & \sigma_{{pe}_{i}{pe}_{j}}\\ 0 & \sigma_{{pe}_{j}}^{2} \end{array}\right]$, and $\left[\begin{array}{c} e_{i}\\ e_{j} \end{array}\right]\sim N(0,I⨂\left[\begin{array}{cc} \sigma_{e_{i}}^{2} & \sigma_{e_{i}e_{j}}\\ 0 & \sigma_{e_{j}}^{2} \end{array}\right]$. $\sigma_{a_{i}a_{j}}$, $\sigma_{{pat}_{i}{pat}_{j}}$, $\sigma_{{pe}_{i}{pe}_{j}}$, and $\sigma_{e_{i}e_{j}}$ are the additive genetic covariance, permanent environmental covariance, and residual covariance between traits i and j, respectively.

According to an expansion of the Taylor series [23], the heritability (h^2^), repeatability (r^2^), and genetic correlations (R) for the IVEP traits were calculated using the following:$$h^{2}=\frac{\sigma_{a}^{2}}{\sigma_{a}^{2}+\sigma_{pat}^{2}+\sigma_{pe}^{2}+\sigma_{e}^{2}}$$
$$r^{2}=\frac{\sigma_{a}^{2}+\sigma_{pe}^{2}}{\sigma_{a}^{2}+\sigma_{pat}^{2}+\sigma_{pe}^{2}+\sigma_{e}^{2}}$$
$$R=\frac{cov(\sigma_{a_{i}}{,\sigma }_{a_{j}})}{\sqrt{\sigma_{a_{i}}*\sigma_{a_{j}}}}$$
where $\sigma_{a_{i}}^{2}$ and $\sigma_{a_{j}}^{2}$ are the additive genetic variance of traits i and j, respectively; $\sigma_{a_{i}a_{j}}$ is the additive genetic covariance between trait i and j.

## 3. Results

### 3.1. Descriptive Statistics

The descriptive statistics for five IVEP traits are presented in Table 1. On the observed scale, the NCOC, NCLV, and NGE had averages of 12.43 ± 7.57, 6.25 ± 4.60, and 2.38 ± 2.63. NGE had the largest coefficient of variation (110.32%), and the CVs of NCOC and NCLV were similar (60.92–73.61%). The CVs of NCOC, NCLV, and NGE on the log-transformed scale were greatly narrowed down (21.82–78.08%). In addition, the average proportion of NCLV and NGE to NCOC were 50.28% and 19.15%, respectively.

The distribution of NCOC, NCLV, and NGE per OPU-IVEP is shown in Figure 1A,B. The number of cumulus–oocyte complexes was between 3 and 22 (89.51%), while the majority number of cleaved embryos greatly decreased to 0–14 (94.16%), and most of donors had only less than two grade I embryos (63.10%).

The distribution of donors with different OPU frequencies is shown in Figure 2. Most donors underwent OPU more than five times (85.35%), and the average OPU frequency was 8.4 times.

The change trends of NCOC, NGE, and PGE in different months are shown in Figure 3. The NCOC presented by OPU had no intensive fluctuation. There were significant decreases in both NGE (3.55 to 1.24) and PGE (24.42% to 10.08%) from June to August, and embryo development was sensitive to the ambient temperature. Furthermore, the PGE started to increase immediately when the donors avoided the heat stress in summer.

The change trends of NCOC, NGE, and PGE per OPU operation, along with the impacts of donor age, are presented in Figure 4. Compared to the donors in the high-frequency group, lower NCOC, NGE, and PGE (Figure 4A,B) were observed in the donors of the low-frequency group. Among younger donors (less than 12 months), the NCOC presented a decreasing trend with the increase in age in both the low- and high-frequency groups. The NGE and PGE generally presented increasing trends with increasing donor age in both the low- and high-frequency groups, but there was a large fluctuation for PGE in the low-frequency group.

### 3.2. Genetic Parameters

Estimates regarding heritability and repeatability for the IVEP traits are shown in Table 2. Moderate to high heritability ranging from 0.13 to 0.34 was observed for count IVEP traits (NCOC, NCLV, and NGE) on the observed scale and the log-transformed scale. Compared to the estimates on the observed scale, slightly lower estimates of heritability were observed for NCLV and NGE on the log-transformed scale. The repeatability of the count IVEP traits ranged from 0.25 to 0.46 on the observed scale, and slightly lower estimates of repeatability were obtained on the log-transformed scale.

The estimates of heritability and repeatability for PCLV and PGE are also shown in Table 2. There were low heritabilities for PCLV and PGE, ranging from 0.06 to 0.10. Interestingly, it was observed that PGE had a higher heritability than that of PCLV.

The genetic correlations among five IVEP traits on the observed scale and log-transformed scale are shown in Figure 5, and detailed estimates are showed in Appendix A. In general, the correlations between the IVEP traits were slightly lower on the log-transformed scale. The highest genetic correlation (0.95 and 0.94) was observed to be between NCOC and NCLV. There was a high correlation between NCLV and NGE (0.77 and 0.69), and this correlation is higher than the estimate of the correlation between NCOC and NGE (0.67 and 0.61). In addition, the genetic correlation between PCLV and PGE was moderate (0.42 on both scales). Interestingly, it was observed that the genetic correlations between count IVEP traits (NCOC, NCLE, and NGE) and percentage IVEP traits (PCLV and PGE) differed greatly. For example, both PCLV and PGE usually had moderate to high genetic correlations with their corresponding count traits (NCLV and NGE), while the other correlation between count traits and percentage traits was low (−0.16–0.05).

The average number and EBV values of the best and worst donors for NCOC are shown in Table 3. The mean EBV of NCOC is −0.012, far less than the best 10% donors, and the difference between the best and worst donors is 0.152. The mean value of NCOC presents a similar difference between the best and worst donors. On average, the best donors had 24.02 oocytes, while the worst donors had, on average, only 2.6 oocytes.

The same trend was found for NGE and PGE. The best donors produced, on average, 3.42 grade I embryos, and the percentage of NGE to NCOC was 30.54%. In contrast, the worst donors produced only 0.36 grade I embryos, and the percentage was 3.46%.

## 4. Discussion

In this study, the averages of NCOC, NCLV, and NGE were 12.43 ± 7.57, 6.25 ± 4.60, and 2.38 ± 2.63, respectively. A similar study on Holstein cattle was conducted by Merton et al. [19], and they found averages for NCOC, NCLV, and NGE of 7.8, 4.4, and 1.1, respectively. Parker et al. [2] reported the highest NCOC and NCLV in Holstein cattle, showing an average of 20.95 for NCOC and 15.81 for NCLV. However, the average of NGE (2.34) was close to that reported in the current study. The averages of NCOC in Japanese black cows and Guzerá cattle (Bos indicus) were relatively consistent (15.19 and 15.60), while the averages of NGE or transferable embryos were found to be higher than that in Holstein cattle, which were reported to range from 5.96 to 6.40 [14,18,24].

It is well known that heat stress has a significant impact on cows, including on their follicular growth and development [25]. In this study, a rapid decline in NCOC, NGE, and PGE appeared from June to August. Although PGE showed an upturn from September, NCOC and NGE remained low. Li et al. [8] reported similar results in Holstein and Luxi cattle, and the averages of NCOC and PGE were both significantly low in summer and autumn due to continuous heat stress. The quantity and quality of follicles and oocytes in cows were also found to be significantly low under heat stress [6,26]. The above studies can be partly explained by the consistent decrease in androstenedione in the follicles of cows under heat stress [26]. It has also been found that the early development of follicles is characterized by a decrease in androgen production by thecal cells [27]. Exposure to summer heat stress causes significant a decrease in androstenedione production by dominant follicles and continues to affect follicles in the autumn [28]. In addition, follicular formation is probably postponed due to delayed luteolysis and reduced estradiol levels when exposed to heat stress, which lead to worse OPU outcomes [29].

Due to the artificial selection, the donors with good performance were given priority for more opportunities of OPU operation. In this study, donors were divided into a high-frequency group (OPU frequency of 9 times or more) and low-frequency group (OPU frequency of less than 9 times) according to the frequency of OPU-IVEP. Donors in the high-frequency group usually had consistent records at every month and age, whereas donors in the low-frequency group were unable to achieve the same level of consistency. Considering the possibility of different trends appearing during aging, it was necessary to analyze the effect of age in the two groups separately. Interestingly, the current study demonstrated that artificial selection had positive effects on the outcome of OPU-IVEP. Furthermore, in both groups, the elder donors always had higher PGE than the younger donors, which is similar to previous findings in buffalo [7] and Holstein populations [12]. Based on the above findings, the attainment of high embryo production efficiency can be expected when conducting the operation on donors in the form of heifers over the age of 15 months.

This study firstly estimated the genetic parameters of IVEP traits in Chinese Holstein cattle. In general, low to moderate heritability (0.007–0.323) and repeatability (0.015–0.627) were found for the count IVEP traits (NCOC, NCLV, and NGE) in Holstein, dairy Gir, and Guzerá populations [2,14,17,18,19]. The estimated heritability for NCOC in this study was slightly higher than the estimates (0.177–0.323) reported by Parker et al. [2], Perez et al. [14], Vizoná et al. [17], and Merton et al. [19]. The estimates of heritability for NCLV in the current study ranged from 0.18 to 0.24—similar to but slightly higher than the estimates reported in previous studies (0.131–0.190) [2,14,18,19]. The estimates of heritability for NGE (0.13–0.16) obtained in the current study, as expected, fit into the range reported in the literature (0.03–0.21) [2,14,18,19]. The paternal random effects on NCLV were relatively low, but the effects on NGE were considerable because the paternal variance component is approximately one-third of the additive genetic variance. The paternal effects present even higher proportions for percentage traits which were nearly 1.5 times for PCLV and 50% for PGE. This result is consistent with that reported by Vizoná et al. [17] in their study on dairy Gir cattle. Vizoná et al. found that the paternal genetic effects for NEMB were low, but the paternal variance component for PGE was nearly the same as the additive genetic variance. Perez et al. [14] also found an increasing variance component from count traits (NCLV and NEMB) to percentage traits (PCLV and PEMB). Interestingly, in this study, the heritability for NCLV and NGE on the observed and log-transferred scales presented special trends. Perez et al. [18] reported a similar finding, as in their study, the estimates of heritability were higher on the log-transferred scale than that on the observed scale, including for NCLV (0.15 verses 0.13) and the numbers of transferable embryos (0.14 verses 0.11). In contrast, in vivo-derived embryo production traits presented higher estimates of heritability on the observed scale than on the log-transferred scale. For example, Ogawa et al. [24] reported increasing estimates from 0.25 (log-transferred scale) to 0.27 (observed scale) for NGE. Low estimates of heritability (0.06 and 0.10) and repeatability (0.07 and 0.17) were observed for PCLV and PGE in this study, and these values were similarly low in previous studies [14,17,19]. Overall, the IVEP traits are heritable, and the outcomes of OPU-IVEP can be improved through genetic selection [30], even though the percentage IVEP traits can possibly make slow progress in breeding.

Moderate to high genetic correlations were observed among NCOC, NCLV, and NGE, similar to several previous reports [14,17,19]. Previous studies on in vivo embryo production have also reported consistent results. For example, a high genetic correlation (0.741–0.969) was found between the total number of oocytes and the number of transferable embryos in Holstein cattle in [15,31]. However, low to moderate genetic correlations have been observed between certain count IVEP traits (NCOC, NCLV and NGE) and percentage IVEP traits (PCLV and PGE), especially between NCOC and percentage IVEP traits. In previous studies, the genetic correlations between NCOC and PGE or percentage of transferable embryos have presented significantly low genetic correlations (−0.20–0) [14,17,19]. The donors producing more oocytes may not present the same performance in embryo production efficiency. In addition, oocytes and follicular morphology and quality have been shown to greatly impact embryo development. For example, it was found that high-quality oocytes brought about a higher fertilization rate and increased embryo quality in [32]. As reported by Merton [5], a greater number of embryos was produced by Class 1 COCs (30%) compared with Class 2 COCs (19%), and Hasler [33] reported a finding that is consistent with this (29% versus 8%). It is supposed that increased NCOC brings about an increased risk of obtaining low-quality oocytes, which are unable to develop into transferable embryos.

Huge variance within the population was found in this study. The best donors produced, on average, 24.02 oocytes and 3.42 grade I embryos, and the worst donors only produced 2.60 oocytes and 0.36 grade I embryos, and this trend was shown for the percentage of NGE to NCOC. A similar result was presented in another study on Canadian Holstein cattle, wherein it was found that the number of embryos (NE) and viable embryos (VE) via superovulation also had large variance. The NE and VE values for the best 10% donors were 15.61 and 13.40, respectively, while the numbers were less than 5 for the worst 10% donors [15]. Vizoná et al. [17] also reported a great level of variation in IVEP traits. The number of grade I oocytes, number of viable oocytes, and percentage of grade I oocytes were observed to observe the differences between the daughters of the top 10% of bulls and the bottom 10% of bulls. It is apparent that high variation existed among the analyzed groups, and this variation remains to be considered in breeding. Selectin the top donors will significantly improve the outcomes of in vitro embryo production and profitability.

## 5. Conclusions

This is the first study of its kind to investigate the population characteristics of in vitro embryo production traits in Chinese Holstein cattle. In this study, in vitro embryo production (IVEP) traits were found to be significantly influenced by season and donor age. Donors producing in vitro embryos should be protected from heat stress in order to attain ideal outcomes. Elder donors are more likely to yield high efficiency in embryo production and produce good embryos. In vitro embryo production traits have low to high heritability, and genetic selection would improve the economic merit of in vitro embryo production. The count IVEP traits (NCOC, NCLV, and NGE) and percentage IVEP traits (PCLV, PGE) reflect different aspects of in vitro embryo production, including donor performance in oocyte recruitment, oocyte quality, and so on. When embryo production needs to be carried out on a large scale, these traits should be considered in genetic selection for dairy cattle.

## Figures and Tables

**Figure 1 animals-13-03539-f001:**
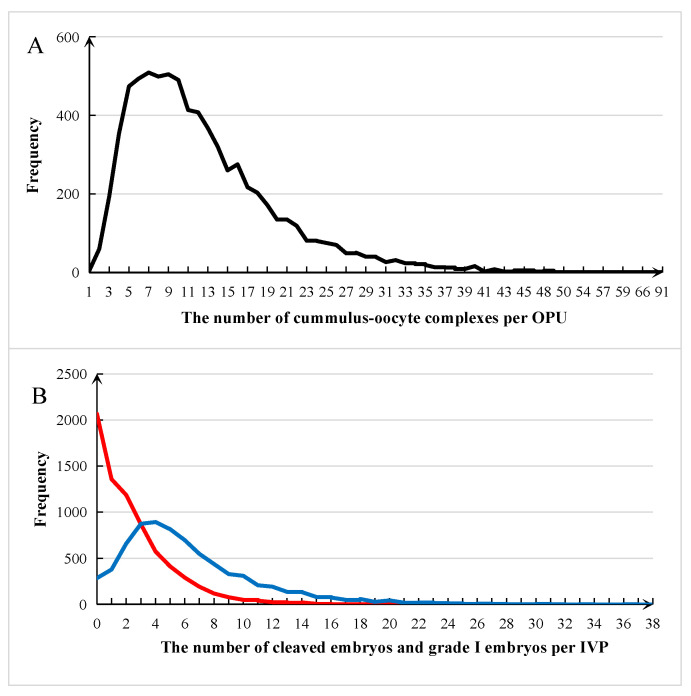
Distributions for the number of cumulus–oocyte complexes, cleaved embryos, and grade I embryos per ovum pick up in Chinese Holstein cattle. (**A**) The number of cumulus–oocyte complexes (black line); (**B**) the number of cleaved embryos (blue line) and the number of grade I embryos (red line).

**Figure 2 animals-13-03539-f002:**
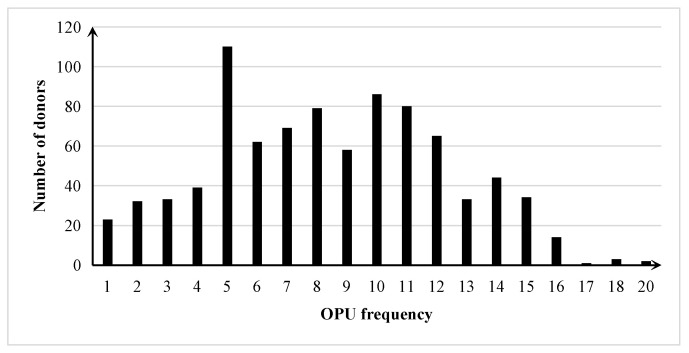
The distributions for donors with different frequencies of ovum pick up (OPU) in Chinese Holstein cattle.

**Figure 3 animals-13-03539-f003:**
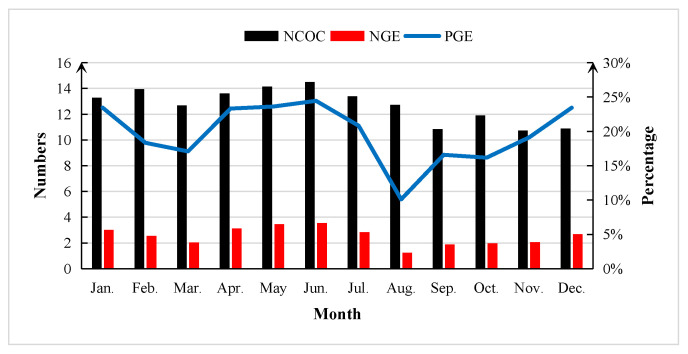
The change trend of average NCOC, NGE, and PGE over the year in Chinese Holstein cattle. NCOC = the number of cumulus–oocyte complexes; NGE = number of grade I embryos; PGE = percentage of grade I embryos.

**Figure 4 animals-13-03539-f004:**
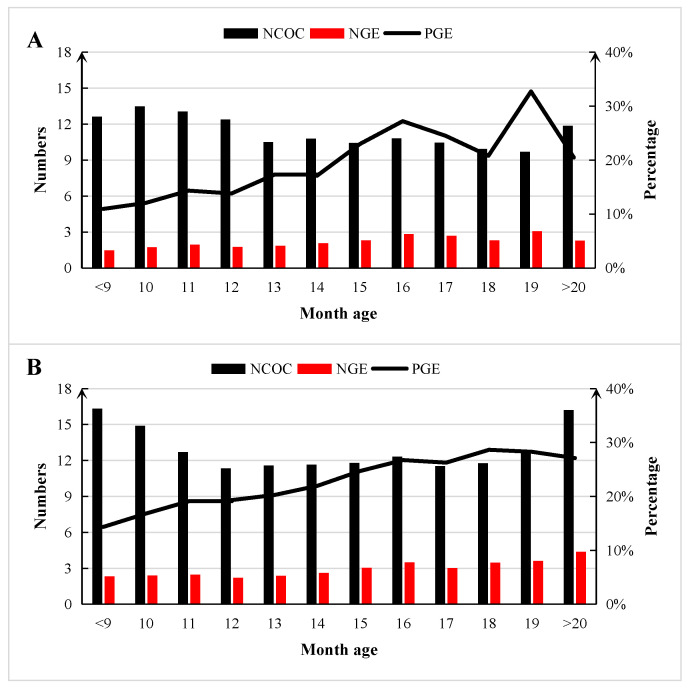
The impacts of donor age on NCOC, NGE, and PGE in Chinese Holstein cattle. (**A**) The average NCOC, NGE, and PGE of donors in the low-frequency group (an OPU frequency of less than 9 times per donor). (**B**) The average NCOC, NGE, and PGE of donors in the high-frequency group (an OPU frequency of 9 or more). NCOC = the number of cumulus–oocyte complexes; NGE = number of grade I embryos; PGE = percentage of grade I embryos.

**Figure 5 animals-13-03539-f005:**
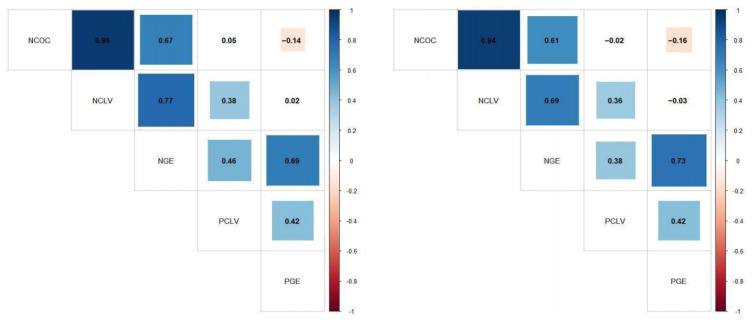
Genetic correlations for in vitro embryo production traits on the observed scale (**left**) and the log-transformed scale (**right**) in Chinese Holstein cattle. NCOC = number of cumulus–oocyte complexes; NCLV = number of cleaved oocytes; NGE = number of grade I embryos; PCLV = the proportion of NCLV to NCOC; PGE = the proportion of NGE to NCOC.

**Table 1 animals-13-03539-t001:** Descriptive statistics of in vitro embryo production traits in Chinese Holstein cattle.

Scale	Trait	Mean	SD	CV	MIN	MAX	No. of Zeros
Observed scale	NCOC	12.43	7.57	60.92%	0	91	0
	NCLV	6.25	4.60	73.61%	0	38	284
	NGE	2.38	2.63	110.32%	0	23	2067
Log-transformed scale	NLCOC	1.07	0.23	21.82%	0.30	1.96	0
	NLCLV	0.77	0.29	37.66%	0	1.29	284
	NLGE	0.41	0.32	78.05%	0	1.38	2067
Observed scale	PCLV	0.50	0.21	41.35%	0	1	284
	PGE	0.19	0.18	94.34%	0	1	2067

NCOC = the number of cumulus–oocyte complexes; NCLV = the number of cleaved embryos; NGE = the number of grade I embryos; PCLV = the proportion of NCLV to NCOC; PGE = the proportion of NGE to NCOC.

**Table 2 animals-13-03539-t002:** Estimates of heritability and repeatability for in vitro embryo production traits in Chinese Holstein cattle.

Item ^1^	Observed Scale				Log-Transformed Scale			
	NCOC ^2^	NCLV	NGE	PCLV	PGE	NCOC	NCLV	NGE
$\sigma_{a}^{2}$	18.62	4.93	1.07	0.0024	0.0032	0.018	0.016	0.013
$\sigma_{pat}^{2}$	0	0.35	0.32	0.0034	0.0018	0	0.011	0.008
$\sigma_{pe}^{2}$	7.37	2.08	0.61	0.0004	0.0016	0.006	0.005	0.009
$\sigma_{e}^{2}$	30.12	13.39	4.69	0.0363	0.0241	0.029	0.057	0.071
$h^{2}$	0.33 (0.06)	0.24 (0.05)	0.16 (0.04)	0.06 (0.02)	0.10 (0.03)	0.34 (0.06)	0.18 (0.04)	0.13 (0.03)
$r^{2}$	0.46	0.34	0.25	0.07	0.17	0.45	0.23	0.22

^1^ $\sigma_{d}^{2}$ = the additive genetic variance; $\sigma_{pat}^{2}$ = the paternal random effects variance; $\sigma_{pe}^{2}$ = the permanent environment variance; $\sigma_{e}^{2}$ = residual variance; h^2^ = heritability; r^2^ = repeatability; ^2^ NCOC = the number of cumulus–oocyte complexes; NCLV = number of cleaved oocytes; NGE = number of grade I embryos; PCLV = the proportion of NCLV to NCOC; PGE = the proportion of NGE to NCOC.

**Table 3 animals-13-03539-t003:** Average number and EBV values of the best (10% quantile) and the worst (90% quantile) donors for NCOC, NGE, and PGE from a univariate linear animal model on the log-transformed scale.

Traits	Item	Mean	10% Quantile	90% Quantile
NCOC	EBV	−0.012	0.174	0.022
	Phenotype	12.88	24.02	2.60
NGE	EBV	0.015	0.158	−0.108
	Phenotype	1.38	3.42	0.36
PGE	EBV	0.014	0.106	−0.057
	Phenotype	11.98%	30.54%	3.46%

NCOC = number of cumulus–oocyte complexes; NGE = number of grade I embryos; PGE = the proportion of NGE to NCOC.

## Data Availability

The data are not publicly available because they contain information that has not yet been published.

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
