# Peer review of "Phenotypic and Genetic Analyses of In Vitro Embryo Production Traits in Chinese Holstein Cattle"

_animals, 2023, doi:10.3390/ani13223539_

Round 1

Reviewer 1 Report

Comments and Suggestions for Authors

The authors present a study with aims to analyse the effects influencing the production efficiency of OPU-IVEP in 867 Chinese Holstein heifers. Traits studied include the number of cumulus-oocyte complexes (NCOC), the number of cleaved embryos (NCLV), the number of grade I embryos (NGE), the proportion of NCLV to NCOC (PCLV) and NGE to NCOC (PGE).

Comments:

Line 104: do not understand what objectives the authors have.

Please clarify what you mean by population characteristics. Which novel trait should be developed?

Can you please outline the design donor by recipient. The design may influence the outcomes of your study. Please also clarify possible confunding among effects in the model used.

Model: additive genetic and permanent environmental effects refer to donors.

The maternal effects (additive genetic and  permanent environment) of the recipients are not considered.

The embryonic effects are not in the model and the paternal genetic effects are omitted.

Service sire: should this be a permanent environmental effect at least or even in addition an additive genetic effect.

There is need to explain the model in more detail.

In addition, the authors have to screen literature for other animal effects than just the donor effects.

I guess the embryo itself should have also effects on the outcome.

Dams used as donors may be selected according to their genetic potential for milk production or even a total merit index.

Can you provide further data on this aspect.

The model part needs extensive revision and also a more thorough review of previous reports.

Calculation of genetic correlations looks strange. We have a covariance component which is standardized on the square root of two variance components: cov (a1-a2) / sqrt ((var a1) (var (a2)) .

sigma d= unusual for additive genetic variance, please use the subscript "a".

Heritability of what: needs discussion. It looks like a maternal heritability (donor). The animal should be the embryo. Donor is the biological dam.

Selection for economic merit of in-vitro embryo production may be discussed, but without the genetic potential for the future cow or bull this would help nothing.

Please improve your discussion and conclusions. There are many issues which need consideration in the analyses and discussion and conclusions.

Comments on the Quality of English Language

No comments

Reviewer 2 Report

Comments and Suggestions for Authors

This is an interesting manuscript aimed to study five in-vitro embryo production traits in Chinese Holstein cattle in order to estimate phenotypic and genetic parameters. I only suggest considering some general comments and some minor grammar corrections:

-       Line 36: I suggest including in the objective the variables season and age, as they are reported in results.

-       Line 90: Replace “Dairy” by “dairy”.

-       Line 91: Replace the comma by a period after the last square bracket.

-       Line 93: Replace “variable” by “variation”.

-       Line 97: Replace “essential” by “needed”.

-       Line 104: The objective is somewhat unclear. Please clarify what do you mean by “characteristics” and what is the “novel traits”.

-       Line 114: How the season in which the cows were worked was classified?

-       Line 142: Please define the studied traits as mentioned in the subtitle.

-       Line 157: Please describe what are the DMU software, as well as its webpage address and the accession date.

-       Line 183: Please mention what was the software used to perform heritability and repeatability analyses, the webpage address and the accession date.

-       Line 183: Replace “calculated” by “were calculated”.

-       Line 202: Replace “Figure” by “Figures”.

-       Line 203: Replace “B” by “1B”.

-       Line 215: Remove “The”.

-       Line 231: Replace “(Figure. 4 A and B)” by “(Figures 4A and 4B)”.

-       Line 265: Replace “Table” by “Tables”.

-       Line 265: Replace “were” by “was”.

-       Line 281: I suggest o include in the first paragraph of the Discussion a brief explanation of the relevance to analyze the five traits involved in the study.

-       Line 363: Was the service sire included in the model to estimate genetic parameters?, this was mentioned in Materials and Methods?

-       Line 368: Please describe what were the “different aspects”.

-       Line 396: Please correct grammar in all references by following the guidelines described in the “Instructions fir Authors”.

Reviewer 3 Report

Comments and Suggestions for Authors

The manuscript “Phenotypic and genetic analyses of in-vitro embryo production traits in Chinese Holstein cattle” aimed to evaluate the phenotypic and genetic characteristics of in-vitro embryo production traits in Chinese Holstein cattle. Unless I judge better, I do not see innovative and technical aspects of this manuscript that will result in publication. The information is based on a sample evaluation of concepts and parameters already known about their influences.

Round 2

Reviewer 1 Report

Comments and Suggestions for Authors

The authors have improved their manuscript and answered all issues very well.

I have no further comments.  The manuscript looks fine.

Typos have to be amended.

Comments on the Quality of English Language

 Some minor English editing seems necessary.

Reviewer 3 Report

Comments and Suggestions for Authors

No comments